# Over 1000-fold enhancement of upconversion luminescence using water-dispersible metal-insulator-metal nanostructures

Ananda Das[1], Chenchen Mao[2], Suehyun Cho[2], Kyoungsik Kim[3] & Wounjhang Park[2,4]

Rare-earth activated upconversion nanoparticles (UCNPs) are receiving renewed attention for use in bioimaging due to their exceptional photostability and low cytotoxicity. Often, these nanoparticles are attached to plasmonic nanostructures to enhance their photoluminescence (PL) emission. However, current wet-chemistry techniques suffer from large inhomogeneity and thus low enhancement is achieved. In this paper, we report lithographically fabricated metal-insulator-metal (MIM) nanostructures that show over 1000-fold enhancement of their PL. We demonstrate the potential for bioimaging applications by dispersing the MIMs into water and imaging bladder cancer cells with them. To our knowledge, our results represent one and two orders of magnitude improvement, respectively, over the best lithographically fabricated structures and colloidal systems in the literature. The large enhancement will allow for bioimaging and therapeutics using lower particle densities or lower excitation power densities, thus increasing the sensitivity and efficacy of such procedures while decreasing potential side effects.

[1] Department of Physics, University of Colorado, Boulder, CO 80309-0390, USA. [2] Department of Electrical, Computer, and Energy Engineering, University of Colorado, Boulder, CO 80309-0425, USA. [3] School of Mechanical Engineering, Yonsei University, 50 Yonsei-ro, Seodaemun-gu, Seoul 03722, Republic of Korea. [4] Materials Science and Energy Engineering, University of Colorado, Boulder, CO 80309, USA. Correspondence and requests for materials should be addressed to W.P. (email: won.park@colorado.edu)

Upconversion luminescence materials combine two or more low energy photons to generate single high energy photons[1–3]. This nonlinear upconversion is most efficiently achieved by energy transfer upconversion or excited state absorption via long-lived intermediate energy levels of the upconversion luminescence material[4]. Unlike the nonlinear susceptibility-based upconversion mechanisms, such as high harmonic generation or optical parametric oscillation, upconversion luminescence can occur with low power excitation and with an incoherent excitation source, and thus has applications in photovoltaics[5–7], displays[8–10], bioimaging[2,11–13], and therapeutics[14–17].

Typical upconversion materials are based on lanthanide ions such as $Tm^{3+}$, $Ho^{3+}$, and $Er^{3+}$, which exhibit efficient excited state absorption[18]. To further improve the upconversion efficiency, additional sensitizer ions can be added. Sensitizers should have strong absorption at wavelengths which match the intermediate energy levels of the activator ions used in upconversion. This way, one can establish an efficient Förster energy transfer from sensitizer to activator, increasing the population of intermediate states and consequently the rate of upconversion transition. A commonly used sensitizer is $Yb^{3+}$. Thanks to its simple two-level energy structure, $Yb^{3+}$ does not suffer from concentration quenching and can therefore be doped much more heavily than most other lanthanide ions. Its excited state energy also closely matches that of the intermediate energy level of $Er^{3+}$ ion, making the $Yb^{3+}$–$Er^{3+}$ co-doped system one of the most efficient upconversion materials. In most host materials, $Yb^{3+}$–$Er^{3+}$ co-doping leads to upconversion of near infrared (NIR) photons at 980 nm to visible photons at the green (510–570 nm) and red (630–680 nm) wavelengths. Fluoride hosts such as $NaYF_4$ are known to be the most efficient due to their low phonon energy, and thus low non-radiative transition rates[1].

Upconversion nanoparticles (UCNPs) are particularly useful for biomedical applications as they have excellent photostability, narrow emissions bands with high color purity, and low cytotoxicity. Additionally, the NIR excitation falls within the biological transparency window allowing for deep penetration into biological tissue. However, a widespread use requires a much improved upconversion efficiency which is fundamentally limited by, among other things, the forbidden nature of the f–f transitions in lanthanide ions[4] and the non-radiative processes mediated by bulk and surface defects[1]. Various methods have been used to increase the emission efficiency of the UCNPs, including manipulating the host material by altering the size or doping levels of the particles[19,20], surface passivation to reduce loss due to surface defects[21–25], and plasmonic enhancement[26–34].

Plasmonic enhancement uses metallic nanostructures which increase the light intensity in the vicinity of the metal surface due to surface plasmon resonances. The UCNP efficiency can be increased by tuning these resonances to either the absorption or the emission wavelengths. It has been found that matching the resonance to the absorption wavelength results in a larger boost in efficiency as the upconverted luminescence intensity depends quadratically on the absorption thanks to the nonlinear nature of the upconversion process[32]. There have been various reports of plasmon-enhanced upconversion both in solution and on lithographically prepared substrates[26–41]. A major limitation of chemically coupling plasmonic nanoparticles with UCNPs is inhomogeneity. Due to the lack of precise control of orientation and distance of the plasmonic nanoparticles with respect to the UCNPs, it is extremely difficult to engineer large luminescence enhancements in self-assembled or chemically prepared nanoclusters. Many have reported either upconversion quenching or no enhancement[35–38]. Others tried to control the distance between UCNPs and plasmonic nanoparticles by inserting spacer layers[39,40] or via DNA conjugation[38,41]. Despite the spacer layer control, the largest solution-based two photon upconversion enhancement factor was 12.4[41]. Lithographic fabrication of nanostructures allows for better control of the geometry of the plasmonic structure, as well as the separation between the plasmonic structures and UCNPs. Therefore, in general, larger upconversion enhancements are reported. Saboktakin et al.[26] have inserted UCNPs inside gold hole arrays and reported an enhancement factor of 35, and Zhang et al.[27] have demonstrated 100 times enhancement of green upconversion luminescence by placing UCNPs near gold disks on top of silicon dioxide pillars. However, the non-scalability of common nanolithography techniques and the inability to produce colloidal solutions make these configurations difficult to use in biological applications.

In this paper, we report over 1000-fold enhancement of $NaYF_4$:$Yb^{3+}$,$Er^{3+}$ UCNPs incorporated in a metal–insulator–metal (MIM) nanostructure. The MIMs are fabricated using laser interference lithography, a scalable technique which results in a large area and high uniformity[32,33,42–45]. We first optimize the geometry to achieve resonance at 980 nm and thus maximize enhancement in absorption. We then conduct steady state and transient PL spectroscopy over a wide range of excitation power densities. Comprehensive electrodynamic simulations and a thorough analysis of the rate equations are performed to quantify the effects of plasmon enhancement and quenching. The MIMs are dispersed into water and large enhancement in solution is demonstrated. In addition to this, physical and chemical stability studies of the dispersed MIMs are performed to further demonstrate the bio-applicability of the nanostructures. Finally, the potential use for biomedical applications is explicitly demonstrated by incubating the MIMs with cancer cells. To our knowledge, this paper reports by far the highest enhancement achieved in a system that can be dispersed in water.

## Results

**Design of plasmonic metal–insulator–metal nanostructure.** The design of the MIM nanostructure is schematically shown in Fig. 1a. Gold is used for the top and bottom metal layers and the insulator is made of a monolayer of $NaYF_4$:$Yb^{3+}$,$Er^{3+}$ nanoparticles. The plasmonic mode excited by this structure results in a highly localized field within the insulator layer, which leads to enhanced absorption and upconversion. The geometry of the structure can be tuned to achieve resonance at various frequencies. Here, we target a resonance at 980 nm corresponding to the energy of absorption transition of the $Yb^{3+}$ sensitizer ions in the UCNPs (see Fig. 1b).

A finite-difference time-domain method is used to simulate the expected absorption enhancement from the structure. Figure 1c shows the simulated field profile of the MIM structure normalized by the incident plane wave excitation. The plasmon mode exhibits a strong field enhancement in the insulator layer, resulting in a large absorption enhancement by the UCNPs. By comparing the local light intensity within the insulator layer of the MIM nanostructures to the local intensity that the UCNPs would experience without the presence of any plasmonic nanostructure, the expected absorption enhancement factor can be extracted. The absorption enhancement factor, $F_{ABS}$, is calculated from simulations using the equation

$$F_{ABS} = \frac{\int |\mathbf{E}|^2 dV}{\int |\mathbf{E_0}|^2 dV} \qquad (1)$$

where $\mathbf{E}$ and $\mathbf{E_0}$ are the electric fields with and without the presence of the plasmonic nanostructure, and the integrals are performed over the volume of the insulator layer.

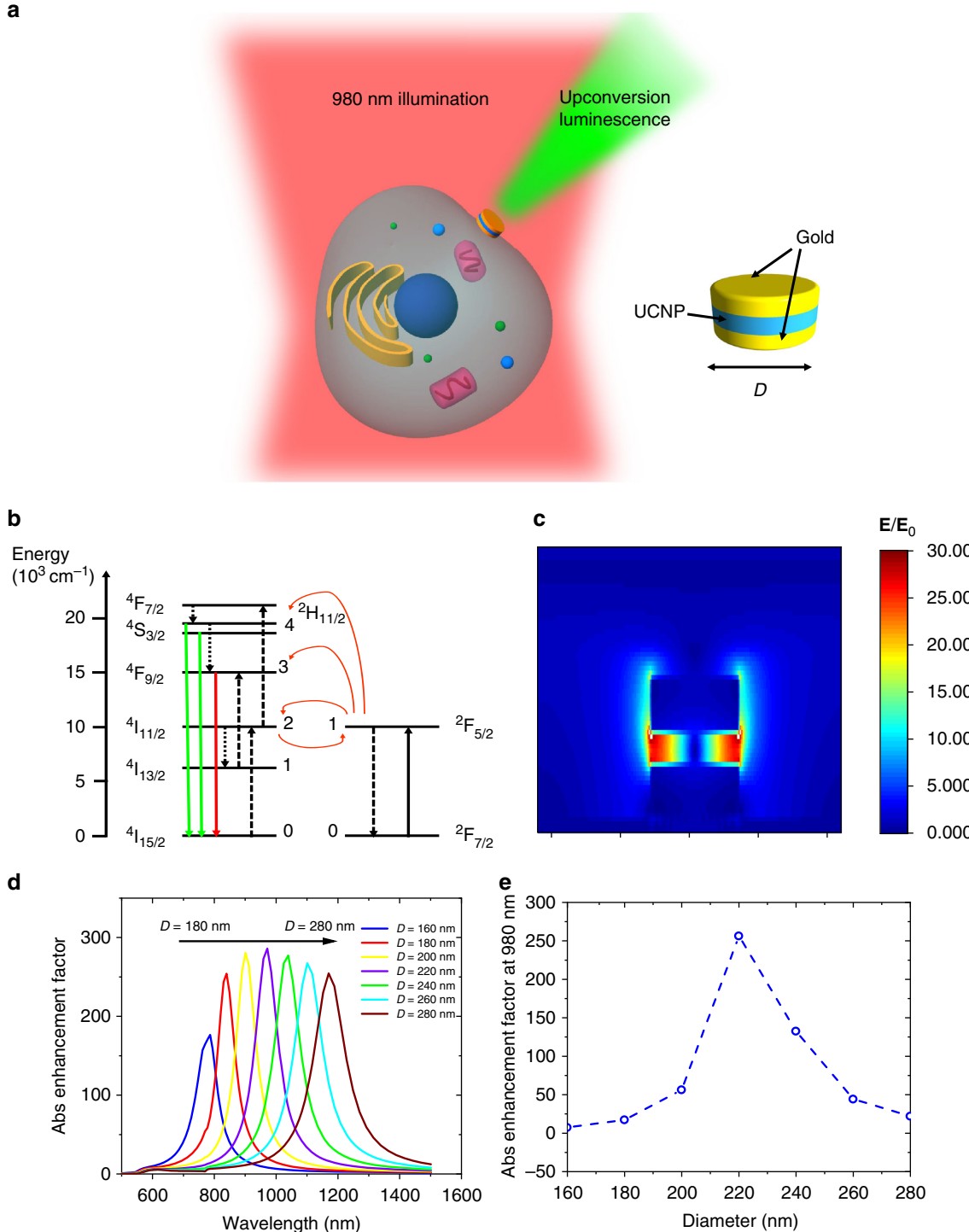

**Fig. 1** Design of dispersible metal–insulator–metal nanostructure. **a** Schematic of metal–insulator–metal (MIM) nanostructure. The top and bottom gold layers squeeze the light into the upconversion nanoparticle (UCNP) insulator layer resulting in a highly localized field that enhances the upconversion process. Poor adhesion between the gold and silicon substrate allows for easy lift off and dispersal into solution. MIMs can be functionalized and attached to cells for bioimaging. **b** Energy levels of donor $Yb^{3+}$ and acceptor $Er^{3+}$ ions relevant to upconversion process. UCNP absorbs 980 nm light and emits upconverted green and red luminescence. **c** Simulated field profile under 980 nm normal incidence plane wave excitation. **E** field is normalized by incident plane wave amplitude $E_0$. **d** Simulated diameter dependence of plasmonic resonance. MIM diameter can be used to tune resonance to any desired wavelength with large absorption enhancement occurring over a wide range of excitation wavelengths. **e** Simulated absorption enhancement factor at 980 nm for various diameters

Figure 1d shows the diameter dependence of the plasmonic resonance wavelength and absorption enhancement factors. As the diameter is increased from 160 to 280 nm, the resonance wavelength can be tuned from 800 to 1200 nm. The resonance can be tuned over a range of desired wavelengths, with absorption enhancement factors of nearly 300 possible over the entire range. By monitoring the 980 nm absorption enhancement at various diameters (see Fig. 1e), we find an optimal diameter of 220 nm

which is easily achievable using laser interference lithography. The simulated absorption enhancements are independent of periodicity, indicating that there are no interactions between the neighboring MIMs and these large enhancements are from the plasmon resonance of individual structures. Therefore, the large enhancements should persist even when the MIMs are lifted off the substrate and dispersed into a solution.

**Fabrication of metal–insulator–metal nanostructure**. The fabrication procedure of the MIMs is illustrated in Fig. 2a. First, a hole array is lithographically patterned onto a photoresist layer to act as a template. Then, a 50 nm thin gold layer is deposited by thermal evaporation. To form the insulator layer, UCNP solution is dropcast onto the samples as described in the Methods section. After the final layer of gold is deposited, the resist template can be removed using acetone. For the preparation of MIM colloidal solutions, the MIMs can be lifted off the substrate using polyvinyl alcohol (PVA) and dispersed in water. Overall, the fabrication process has a few, relatively simple steps and does not require any complicated wet-chemistry techniques usually associated with colloidal plasmon-enhanced UCNPs. The process is readily scalable to a larger area for even higher yield while maintaining excellent uniformity[42].

Figure 2b shows the UCNP particles used for the experiment. The particles have an average diameter of 30 nm. For this study, all PL emissions are compared to a reference sample that was prepared by depositing UCNPs onto an unpatterned silicon wafer using the same dropcast method. A cross-section of this reference sample is shown in Fig. 2c. The dropcast method results in a uniform layer of UCNPs whose thickness can be adjusted by changing the UCNP concentration. Here, we use a reference with a 90 nm thick layer of UCNPs.

Figure 2d–g shows scanning electron microscopy (SEM) images of the samples during various steps in the fabrication process. Figure 2d shows a top-down view of the resist hole array that was used as a template. The hole diameter, and resulting MIM diameter, can be varied easily by changing the exposure time in the lithography step. The samples used in this paper have a periodicity of 730 nm with a hole diameter ranging between 220 and 300 nm. Figure 2e shows a cross-sectional image of the MIM sample after the UCNP deposition step. We can clearly see the bottom layer of gold with a monolayer of UCNPs above it in the two holes shown. The resist hole walls are nonuniform due to an interference pattern formed by the reflection from the substrate during the lithography step. This can be easily mitigated in the future by using an antireflection coating and, as shown by the results, does not adversely affect the performance of the current structures. The toluene that was used as a solvent for the UCNPs degraded the resist and resulted in a slight warping of the holes. This can be observed in Fig. 2e as the top of the holes is slightly narrower than the bottom. Figure 2f shows a top-down view of the MIM sample after the resist was removed. Both the SEM images and PL measurements show that the final array of MIMs is uniform across the sample with PL intensity variations within 10% of the average value over roughly 100,000 MIMs (see Supplementary Figure 1).

An angled SEM image of the final fabricated MIM structure is shown in Fig. 2g. The image shows the distinct layers of gold separated by a layer of UCNPs. However, the top layer of gold is smaller than the bottom layer. This smaller top layer results from the deformation of the hole array due to toluene, as mentioned earlier. Through an analysis of the SEM images of the MIM structure, we see a 30 nm monolayer of UCNPs between the two metal layers, a bottom metal diameter of 250 nm and a top metal diameter of 215 nm. We also observe a few additional UCNPs surrounding the MIM structures. Based on these observations, we built a new model in which the top gold layer is 85% smaller than the bottom gold layer. The simulations show that, despite the slight deviation from the original design, the fabricated structures still support the desired plasmon mode and thus are expected to show high enhancements (see Supplementary Figure 2). The sample shown in Fig. 2g gives the highest enhancement as it is the closest to the optimal geometry suggested by simulations.

**Photoluminescence measurements**. To measure the actual enhancement in upconverted luminescence due to the MIM structure, visible PL is measured under NIR excitation. The UCNPs have visible emission in the green (510–570 nm) and red (630–680 nm) regions. For the rate equation analysis which will be considered in the Discussion, only the emission in the green band is considered. The red emission is also enhanced but the quantitative analysis is much more difficult due to the contributions by three-photon processes[32]. The enhancement is calculated by comparing the integrated PL intensity over the green emission band of the MIMs to that of the reference. The PL intensities are calibrated to account for the difference in the number of UCNPs in the MIM versus the reference. We assume a linear scaling in our calibration, which is valid due to the small absorption cross-section of the UCNPs[32].

Figure 3 shows the results from the PL measurements performed on the MIMs while still attached to the silicon substrate. To achieve the optimal MIM diameter, hole array template diameter is varied from 220 to 300 nm. Figure 3a shows the green PL from MIMs of five different diameters along with the reference sample which is magnified by a factor of 10 to show the relevant features. All samples display the green emission peaks typically associated with the NaYF$_4$:Yb$^{3+}$,Er$^{3+}$ UCNPs. The result suggests that the maximum enhancement occurs at a diameter between 240 and 250 nm, which perfectly agrees with the predictions from simulations (Fig. 3b). We then measure the excitation power dependence of the MIM nanostructure with bottom metal diameter of 250 nm. Figure 3c shows the power dependence of both the MIM and the reference samples. As predicted from theory[32], we see the strong excitation regime, where the PL scales linearly with incident power, and the weak excitation regime, where the PL scales quadratically with incident power. The MIM not only exhibits higher PL intensity but also the strong excitation regime for MIM starts at a lower power density compared to the reference due to the local field enhancement by the plasmon resonance. Finally, we measure the power dependent upconversion enhancement achieved by the MIM nanostructure. Figure 3d shows the enhancement obtained at various incident power densities. We observe the expected plateaus in the strong and weak excitation regimes, corresponding to the ranges of power density where the MIM and reference scale identically with incident power. We measure a PL enhancement factor of 117 in the strong excitation regime and 1198 in the weak excitation regime.

We also perform time-resolved PL measurements on the MIM structures before lifting off and dispersing in water. The equations used to fit the decay measurements are discussed further in the Supplementary Information (see Supplementary Notes 1, 2, 3). By comparing the NIR and green decay rates of the MIM and the reference, we can quantify the effects of metal quenching on the PL intensity. Figure 4a shows the transient green emission curves for the MIM and the reference. We measure a decay rate of $1.00 \times 10^4$ s$^{-1}$ for the MIM and $6.95 \times 10^3$ s$^{-1}$ for the reference. Since the MIM is designed for a plasmon resonance at 980 nm, we expect no Purcell effect for the green emission. The faster MIM decay rate is therefore attributed to the increased non-radiative

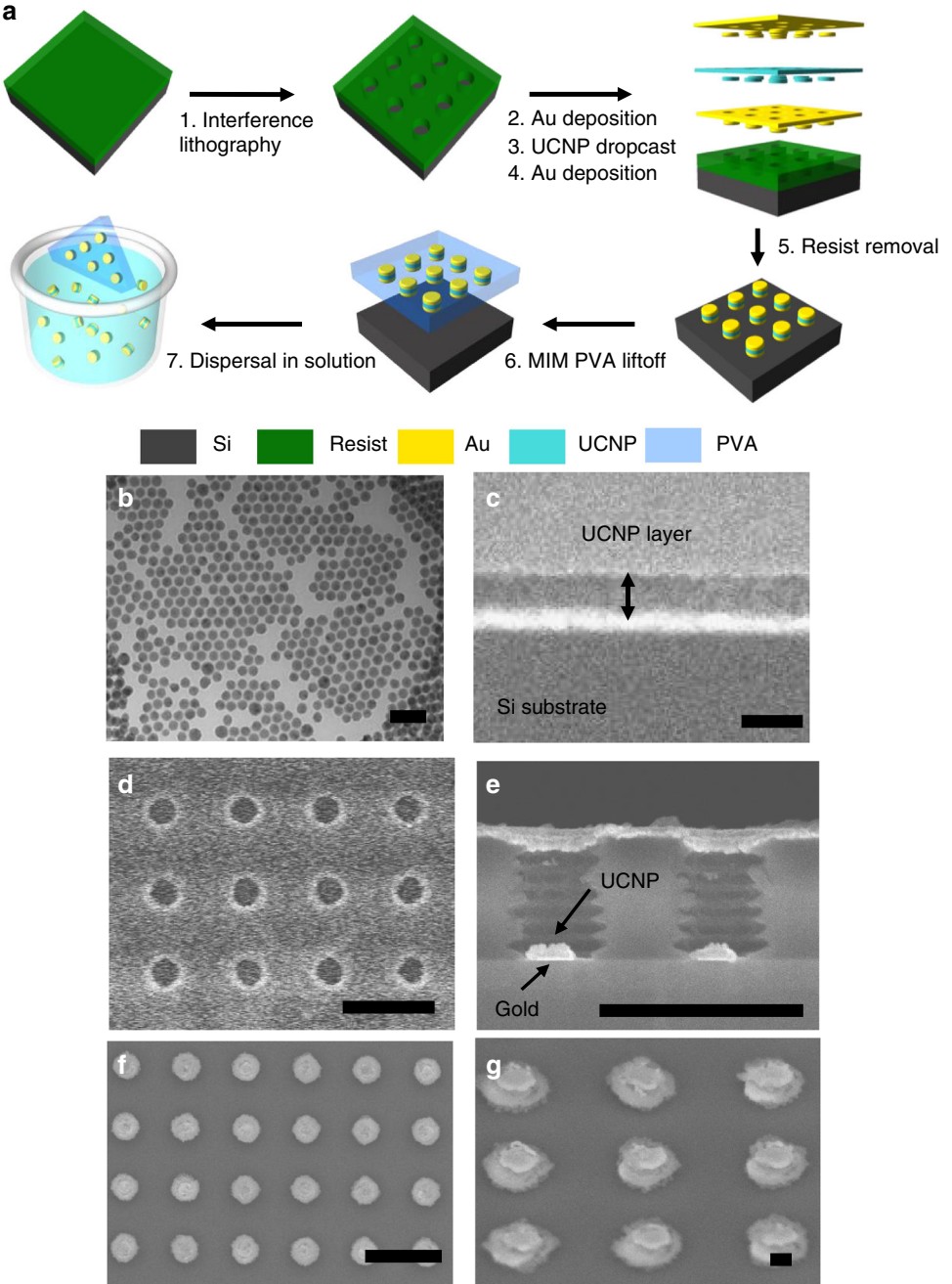

**Fig. 2** Metal–insulator–metal (MIM) fabrication process. **a** Negative resist is exposed using a Lloyds mirror setup to create a hole array template. Metal layers are deposited via thermal evaporation and upconversion nanoparticle (UCNP) insulator layer is deposited using a dropcast method. Once this is done, the resist template is removed using acetone and the resultant array of MIM structures can be removed using polyvinyl alcohol and dissolved into water. **b** Transmission electron microscope (TEM) image of UCNPs used in the experiment. **c** Scanning electron microscope (SEM) image of reference used in the experiment. The dropcast method used to deposit the UCNP layer in our samples results in a uniformly thick layer of UCNPs. **d** Top-down SEM image of resist hole array template. Diameter of holes can be easily varied by varying the resist exposure time. **e** Cross-sectional SEM image of resist hole array template after the UCNP dropcast step. The bottom gold layer and UCNP monolayer is seen within the holes. The top of the holes is warped by the dropcast process and has a smaller diameter than the original hole diameter. **f** Top-down SEM image of MIM sample after the resist is removed. The MIMs are highly uniform with consistently large photoluminescence enhancement visible across the whole sample. **g** Angled SEM image of MIMs after resist removal. Due to warping of the hole diameter during the fabrication process, the top gold diameter is smaller than the bottom gold diameter. Scale bars: **b**, **c** 100 nm; **d**–**f** 1 μm; **g** 100 nm

decay rate due to quenching by the gold layers. By dividing the MIM decay rate by the reference decay rate, we get a visible quenching factor, $Q_{VIS}$, of 1.4. Figure 4b and c show the decay of NIR emission from the reference and the MIM at various power densities. For both samples, higher power densities result in

shorter decay times. This is because at high power densities the intermediate energy levels are highly populated so the energy transfer upconversion rate is increased, resulting in a faster overall decay rate for the NIR emission[33]. In the strong excitation regime, we measure a decay rate of $3.00 \times 10^4 \, \text{s}^{-1}$ for the MIM

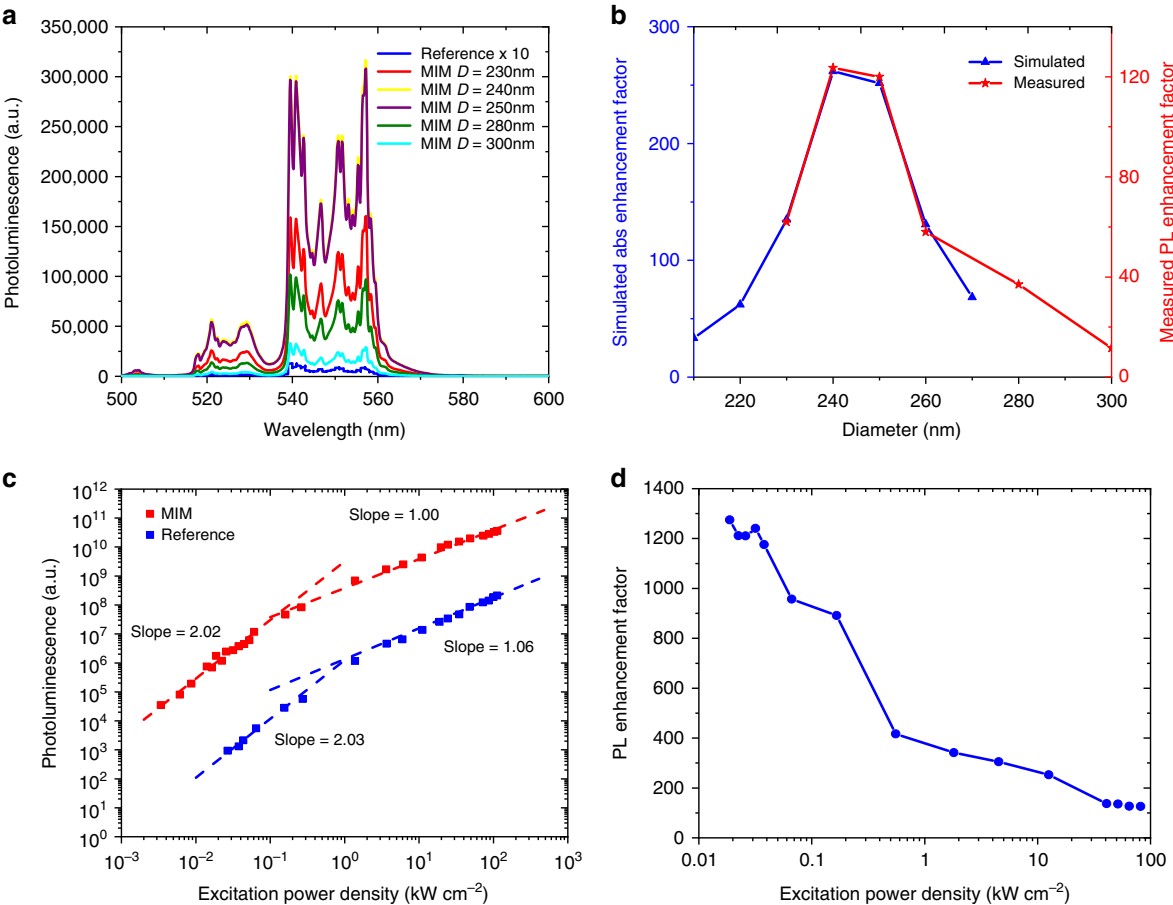

**Fig. 3** Steady state photoluminescence (PL) spectroscopy results. **a** Green emission spectra of reference sample and metal–insulator–metal (MIM) structure of varying diameter under 980 nm excitation. The reference sample emission is magnified by 10 to show the relevant features. **b** Measured versus simulated enhancement at 980 nm for MIM structures of various diameters. The measured enhancement is lower due to metal quenching and roughness but the overall diameter dependence agrees with predictions from simulation. **c** Upconversion luminescence intensity as a function of power density for 250 nm diameter MIM sample and the reference sample. Both samples show linear dependence at strong excitation power densities and quadratic dependence at weak excitation power densities which is expected from the nonlinear upconversion process. **d** 250 nm diameter MIM PL enhancement as a function of power density. At strong and weak excitation power densities, the enhancement plateaus as the PL scales identically for the reference and MIM sample

and $1.26 \times 10^4 \, s^{-1}$ for the reference. In the weak excitation regime, we measure a decay rate of $1.67 \times 10^4 \, s^{-1}$ for the MIM and $5.71 \times 10^3 \, s^{-1}$ for the reference. In the weak excitation regime, upconversion rate is small and only the intrinsic decay process affects the lifetime[33]. Thus, by comparing the NIR decay at weak excitation power densities, we extract a NIR quenching factor, $Q_{NIR}$ of 2.9. From the transient PL measurements, we can also extract the NIR $Er^{3+}$ decay rate $W_{A2}$ (see Supplementary Note 3). $W_{A2}$ is found to be $7.05 \times 10^4 \, s^{-1}$ for the MIM structure and $1.07 \times 10^4 \, s^{-1}$ for the reference.

Finally, the enhancement in PL by the MIMs in solution is confirmed by dispersing the MIMs into water. The MIM sample is compared to a reference of UCNPs in solution and the measured PL is calibrated to account for the difference in concentrations of the two samples. The dispersed MIMs have an enhancement factor of 50 in the strong excitation regime. We could not measure the enhancement in the weak excitation regime as the concentration of MIMs is not high enough to produce a measurable signal.

In summary, we have developed a simple and effective method for large-scale production of MIM nanostructures for plasmon enhancement of luminescence upconversion. The structure can be tuned to any desired wavelength and shows 3 orders of

magnitude of enhancement in upconversion luminescence. By varying the diameter of the hole array template, we identified an optimal diameter of 250 nm corresponding to a 980 nm resonance and obtain an enhancement factor of 117 in the strong excitation limit and 1198 in the weak excitation limit. To quantify the effect of metal quenching, which competes with the plasmon enhancement effect, we have performed transient PL spectroscopy at both the NIR and green wavelengths and obtain quenching factors of 1.4 in the green and 2.9 in the NIR. We have conducted a rigorous analysis of rate equations, taking explicitly into account the non-radiative transition rates due to quenching. The analysis showed that the enhancement in the strong excitation regime is affected only by the quenching of the green luminescence, while in the weak excitation regime both green and NIR quenching must be accounted for. The extremely high measured enhancement factors are found to be consistent with predictions from a theoretical analysis of the UCNP rate equations and electrodynamic modeling of the structure. The observed enhancement exceeds the previously reported upconversion enhancement of lithographically prepared plasmonic structures by an order of magnitude. Furthermore, we have demonstrated that the MIM structures can be made into a colloidal suspension while maintaining the fabricated geometry

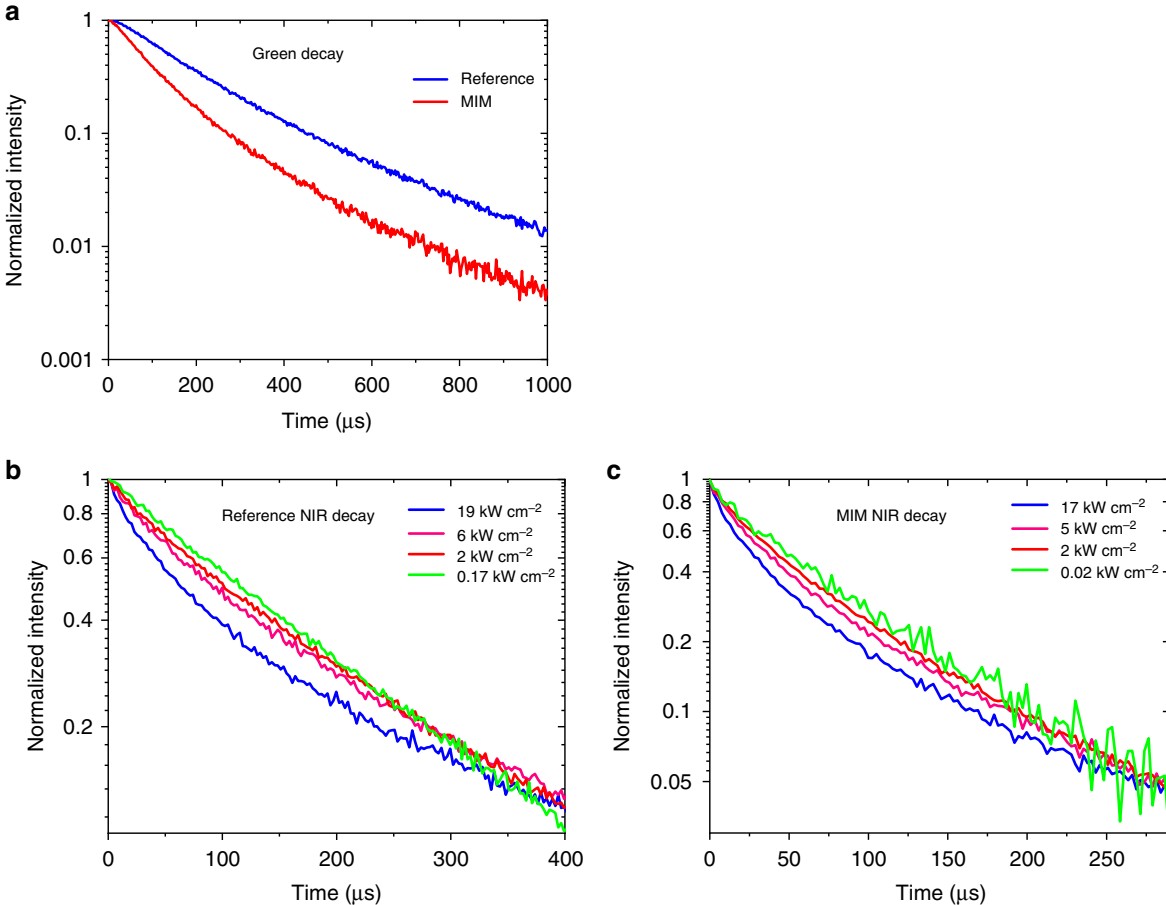

**Fig. 4** Transient photoluminescence spectroscopy results. **a** Green emission decay curve under 980 nm excitation for 250 nm diameter metal–insulator–metal (MIM) sample and reference. The MIM decay is faster due to quenching from the metal layers. **b** Near infrared (NIR) decay curve for reference sample at various excitation power densities. As the excitation power density increases, the NIR decay rate decreases due to an increase in the upconversion process. **c** NIR decay curve for 250 nm MIM sample. The MIM sample shows a similar trend with higher power densities resulting in lower decay rates. At lower power densities, the intrinsic decay process dominates

and therefore high enhancement. Having large PL enhancement in colloidal suspension enables efficient detection and treatment of cancer cells with lower particle densities and lower irradiation power densities, which will ultimately increase the sensitivity and efficacy with minimal potential side-effects.

## Discussion

Figure 1b shows the energy level diagram of the $Yb^{3+}$ and $Er^{3+}$ ions relevant to the energy transfer upconversion process. The rate equations associated with the energy transfer upconversion process can be used to predict the expected emission from the UCNPs. A full analysis of the rate equations is presented in the Supplementary Information (see Supplementary Note 1). The upconverted green photon flux in the strong and weak excitation regimes can be expressed as[32]

$$\Phi_S = \frac{W_{A40}}{W_{A4}} \frac{N_{D0}}{2} \sigma \Phi \qquad (2)$$

$$\Phi_W = \frac{W_{A40}}{W_{A4}} \frac{c_{d4}(c_{Bd2}N_D + W_{A2})N_D^2}{c_{Fd2}N_A \left[ W_{A2} + \frac{W_{D10}(c_{Bd2}N_D + W_{A2})}{c_{Fd2}N_A} \right]^2} (\sigma\Phi)^2 \qquad (3)$$

Here, $N_i$ is the population density in the energy level $i$. The subscripts $D_1$ and $D_0$ refer to the excited ($^2F_{5/2}$) and ground state ($^2F_{7/2}$) levels of donor ($Yb^{3+}$), respectively. $A_4$ and $A_2$ represent the $^4S_{3/2}$ and $^4I_{11/2}$ levels of acceptor ($Er^{3+}$), respectively. $W_{A4}$

and $W_{A40}$ indicate the total and radiative decay rates, respectively, from the initial state $^4S_{3/2}$ to the final state $^4I_{15/2}$. $W_{A2}$ is the total decay rate from the initial state $^4I_{11/2}$ to the final state $^4I_{15/2}$. The energy transfer coefficient $c_{d4}$ represents the Förster energy transfer process between the donor and the acceptor $A_4$ level. For the energy transfer between donor and acceptor $A_2$ level, the additional subscripts $F$ and $B$ in $c_{d2}$ coefficient indicate the forward (donor to acceptor) and backward (acceptor to donor) energy transfers. $N_D$ and $N_A$ are the doping densities of donor and acceptor, respectively. $\sigma$ is the absorption cross-section of the donor ion and $\Phi$ is the incident photon flux. The subscripts S and W indicate the strong and weak excitation regimes, respectively.

The dependence of the green photon flux on the various decay and energy transfer rates allows us to estimate the expected PL enhancement from the MIM structures. Since the same batch of UCNPs was used for the MIM and the reference, both samples have the same doping densities, $N_D$ and $N_A$. The radiative decay rate $W_{A40}$ remains the same for the MIM and reference samples, as there is no resonance in the green so we expect no Purcell enhancement. Finally, the high doping density of our UCNPs results in small separation distances (~1 nm) between the nearest donor and acceptor ion pairs. Plasmon enhancement of the energy transfer rates between the ions is negligible for such small separation distances[34]. Thus, we also assume no plasmonic enhancement of energy transfer coefficients, $c_{Fd2}$, $c_{d4}$, and $c_{Bd2}$.

This leaves us with two competing mechanisms that ultimately determine the upconversion enhancement in the MIM structure. First, the plasmon resonance enhances absorption in the MIM. At the same time, the metal layers in the MIM structure introduce additional non-radiative mechanisms resulting in luminescence quenching. The quenching should make the decay rates $W_{D10}$ and $W_{A4}$ higher for the MIM than in the reference. Given all these considerations and using Eqs. (2) and (3), the expected PL enhancement in the strong and weak excitation regimes can be written as

$$F_S \equiv \frac{\Phi_{S-MIM}}{\Phi_{S-Ref}} = \frac{(\sigma\Phi)_{MIM}}{(\sigma\Phi)_{Ref}} \frac{W_{A4-Ref}}{W_{A4-MIM}} = \frac{F_{ABS}}{Q_{VIS}} \qquad (4)$$

$$F_W = \frac{F_{ABS}^2}{Q_{VIS}} \frac{c_{Bd2}N_D + W_{A2-MIM}}{c_{Bd2}N_D + W_{A2-Ref}} \left( \frac{c_{Fd2}N_A W_{A2-Ref} + W_{D10-Ref}(c_{Bd2}N_D + W_{A2-Ref})}{c_{Fd2}N_A W_{A2-MIM} + W_{D10-MIM}(c_{Bd2}N_D + W_{A2-MIM})} \right)^2 \qquad (5)$$

where $F_{ABS}$ is the absorption enhancement factor, $Q_{VIS} = \frac{W_{A4-MIM}}{W_{A4-Ref}}$ is the quenching factor at green wavelengths and the relevant processes for the plasmonically enhanced MIM versus the reference are distinguished by an additional subscript. Note that the enhancement in the strong regime is only sensitive to quenching at green wavelengths, while the enhancement in the weak regime is also affected by quenching in the NIR. These quenching factors prevent us from achieving a simple quadratic scaling of enhancement in the weak excitation regime and must be accounted for with time-resolved PL measurements.

The enhancement factor in the strong excitation regime, $F_S$, depends only on the absorption enhancement and visible quenching. Comparing the values of $W_{A4}$ between the MIM and the reference from the green decay measurements in Fig. 4a, we find the visible quenching factor $Q_{VIS}$ to be 1.4. This, combined with the measured enhancement factor of 117 in the strong excitation regime (see Fig. 3d), suggests an absorption enhancement factor of 168 from our structure. This is lower than the absorption enhancement factor of 250 predicted by the simulations (see Fig. 3b). The discrepancy stems from the fact that the actual fabricated structure was modeled with a perfectly cylindrical geometry with no roughness in simulations. The measured PL enhancement is still quite high, indicating that the fabricated MIMs exhibit strong local field enhancement despite the unavoidable roughness introduced in fabrication.

The enhancement factor in the weak excitation regime, $F_W$, depends additionally on quenching effects in the NIR. For our analysis, we use a doping density $N_D$ of $1.98 \times 10^{21}$ cm$^{-3}$, a doping density $N_A$ of $2.2 \times 10^{20}$ cm$^{-3}$, a back energy transfer coefficient[46] $c_{Bd2}$ of $1.16 \times 10^{-16}$ cm$^3$ s$^{-1}$ and a forward energy transfer coefficient[46] $c_{Fd2}$ of $1.66 \times 10^{-16}$ cm$^3$ s$^{-1}$. Using these values, and the relevant decay rates and quenching factors reported earlier, Eq. (5) predicts an enhancement factor of 1328 in the weak excitation regime, which agrees well with our measured enhancement factor of 1198.

Briefly summarizing our analysis of the measurements performed on the MIMs still on the silicon substrate, the MIM nanostructure has an absorption enhancement factor of 168, visible quenching factor of 1.4 and NIR quenching factor of 2.9, leading to overall upconversion enhancement factors of 1198 and 117 in the weak and strong excitation regimes, respectively.

To demonstrate potential for use in biomedical applications, the MIMs are removed from the sample and dispersed in water. The MIMs are removed from the sample by flooding the sample with PVA, allowing the PVA to solidify overnight and then removing the portion of PVA covering the MIMs (see Fig. 2a). The PVA film is then dissolved in warm water. Afterwards, the dispersion of the MIM structures is confirmed via transmission

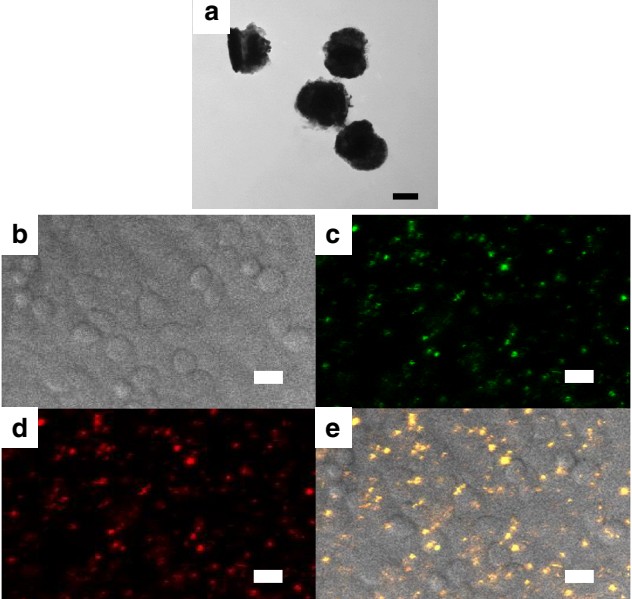

**Fig. 5** Metal–insulator–metal dispersal and incubation into cells. **a** Transmission electron miscroscopy (TEM) images of the metal–insulator–metal (MIM) structures after dispersal in water. The well-defined metal and insulator layers in the image demonstrate that the structure is robust and maintains structural integrity through the lift off process. **b**–**e** Confocal laser scanning microscopy images of T24T bladder cancer cells incubated with MIM structures. **b** Brightfield image of cells, **c** Green, and **d** red upconverion fluorescence collected upon 980 nm excitation show characteristic UCNP signal. **e** Overlay of green and red upconversion PL signals with brightfield. Scale bar: **a** 100 nm; **b**–**e** 20 μm

electron microscopy (TEM) images (see Fig. 5a). As seen from the TEM image, the MIM structures are robust enough to withstand the lift-off process and maintain structural integrity when dispersed in water. This simple lift off method allows for the facile dispersion of a large number of the MIM structures into water which is crucial for the further functionalization steps and subsequent cell targeting associated with many bioimaging and therapeutic techniques.

To predict the expected enhancement factor of the MIMs once dispersed, we simulate the structure under all possible excitation angles and polarizations. Due to the cylindrical symmetry of the structure, we only have to consider excitation along the $z$ axis and excitation at two polarizations along the radial direction. We find that, in addition to the large enhancement demonstrated by excitation along the $z$ axis, enhancement also occurs for one of the two polarizations along the radial direction (see Supplementary Figure 3). Averaging over all angles and polarizations, we predict an enhancement that is about 41% of the enhancement measured from the structures on the silicon substrate. The dispersed MIMs have an enhancement factor of 50 in the strong excitation regime which is 43% of the enhancement factor of 117 that is measured for the structures still on the substrate and agrees well with predictions. As seen in Fig. 3d, this enhancement should become even larger in the weak excitation regime where the nonlinear dependence on incident power density can be exploited.

We also study the physical, chemical, and thermal stability of the dispersed MIM structures (see Supplementary Figures 4, 5). The colloidal stability of the MIM nanostructures in solution is investigated by monitoring dynamic light scattering (DLS) measurements of the MIM solution over time. Sensitivity to the solution's pH value is also tested by preparing three buffer

solutions with pH values of 4, 7, and 10. The DLS measurements show that the particles are stable for up to 8 h in all three buffer solutions (see Supplementary Figure 5). A final measurement is performed 24 h after dispersal but by this point the sample is too polydisperse to perform DLS analysis suggesting the MIMs have broken apart into their constituent layers. This is confirmed with TEM images of the dispersed samples which show structural integrity up to 12 h and degradation at 24 h.

The as-created MIM structures do not have any functional groups or charges on the surface that allows dispersion in water for a long period of time. Based on our TEM and DLS studies, it can be seen that the MIM structures are stable in water for at least 12 h under varying pH conditions. This grants sufficient time for subsequent surface modification for high stability. For example, the MIM structures can be coated with monofunctional thiol-polyethylene glycol (mPEG-SH) that will allow long-term colloidal stability in water. For improved structural integrity, a silica coating can be applied. These coating processes are typically overnight reaction processes[47] and thus we believe they can be done with high efficiency.

As a proof of principle test to show the feasibility for future biomedical applications, the MIM solutions are used to image cancer cells. Figure 5b shows the brightfield image of the T24T cells incubated with the MIM structures. Bright green and red upconversion luminescence is observed under 980 nm excitation, as shown in Fig. 5c, d, respectively. As Fig. 5c, d, and the overlay of green and red PL (resulting in yellow) with the brightfield image in Fig. 5e demonstrate, the upconversion PL from the MIM structures is easily detectable against a very dark background with no autofluorescence, producing crisp, high-contrast fluorescence images. To assess the effectiveness of the MIMs as an imaging probe, we compare with fluorescence images of cells treated with antibody conjugated UCNP and gold nanorod clusters (see Supplementary Figure 6). The images are taken with the same microscope under similar illumination and detection conditions. MIMs are shown to provide a similar level of brightness with 3 orders of magnitude lower concentrations. We also study the effects of neighboring MIMs as a function of the spacing between the MIMs (see Supplementary Figure 7). Our simulations show that separation distances larger than 500 nm are sufficient to avoid any reduction in enhancement due to nearby MIMs. Thus, our results demonstrate that MIMs can easily replace existing water-soluble UCNPs while drastically reducing the total number of nanoparticles needed for high-quality cell imaging.

The extremely high upconversion enhancement and uniformity of MIM structures offer distinct advantages in biological applications. Most solution-based coupling of UCNPs with plasmonic structures have shown either quenching[35–37] or very low enhancement[16,17,28,39,41]. This is due to the large inhomogeneity which is difficult to avoid in colloidal nanostructures and also to the difficulty in producing complex nanoscale geometries. Thus far, the largest solution-based two-photon upconversion enhancement reported was 12.4[41]. The dispersed MIM structures reported in this paper show 4 times higher enhancement than this at high power densities and an order of magnitude improvement can be achieved by going to lower power densities.

Furthermore, the MIM structures can be easily modified via well-established surface modification techniques such as poly (ethylene glycol) (PEG) coating[48], poly electrolyte layer-by-layer assembly[49], DNA modification[50], and silica coating[51]. These surface modification techniques can be used not only to improve biostability of the MIMs in vivo but also to allow for further functionalization. For example, through PEG coating, we can increase the circulation time of the MIM structures in vivo[37,47] and also allow specific targeting of the MIM structures to cancer cells[47]. These surface modifications will enable a wide range of biomedical applications of the MIM structures. Finally, using the NIR plasmon resonance of the MIM structures, we envision various treatment modalities such as photothermal therapy[47], optoporation[52], and selective photodynamic therapy[16,17].

## Methods

**Synthesis of upconversion nanoparticles**. UCNPs were synthesized using the thermal decomposition method[53]. In short, a total of 0.0193 mol of lanthanide precursors ($YCl_3$, $YbCl_3$, and $ErCl_3$) were dissolved in 36 mL of 1-octadecene and 6 mL of oleic acid at 160 °C for 10 min then cooled to room temperature. $NH_4F$ and NaOH were separately dissolved in methanol, added to the lanthanide mixture, and vigorously stirred for 30 min. Then, the mixture was heated to 100 °C and degassed for 20 min. Finally, the mixture was heated up to 310 °C under argon protection for 1 h. Once the UCNP mixture cooled to room temperature, it was washed through centrifugation with water and ethanol. Finally, UCNPs were dried in an oven overnight, redispersed in toluene and filtered with a 0.22 μm PTFE syringe filter. All UCNPs used in this work have doping densities of 2% $Er^{3+}$ and 18% $Yb^{3+}$.

**Fabrication of photoresist hole array**. A negative photoresist (NR9-1000, Futurrex) was spun onto silicon wafers for 1 min at 8000 rpm to create a 750 nm thick resist layer. The wafers were then baked at 150 °C for 1 min and then exposed to a 325 nm HeCd laser (Kimmon). A home-built Lloyd's mirror setup was used to create an interference pattern on the resist layer. A double exposure with a 90° sample rotation in between creates a hole array in the resist. The wafers were then baked at 100 °C for 1 min and then immersed in developer solution (RD-6, Futurrex). Varying exposure conditions resulted in a set of hole arrays with diameters between 220 and 300 nm.

**Fabrication of metal–insulator–metal nanostructure**. Once the hole arrays were made, a 50 nm thick gold layer was thermally evaporated onto the samples, with a very thin of titanium added on top to help with adhesion. A 30 nm UCNP monolayer was added by dropcasting 20 μL of 2 mg mL$^{-1}$ UCNP solution onto the sample substrate. To make the samples more resistant to the toluene, which is used as a solvent in the UCNP solution, the samples were hard baked at 100 °C for 5 min prior to UCNP dropcasting. A final layer of 50 nm thick gold layer, along with a thin Ti layer, was thermally evaporated onto the sample. To remove the resist, the samples were left in warm acetone overnight and then sonicated. Samples were plasma ashed to remove any remaining resist. Finally, the MIM nanostructures were lifted off using PVA and dispersed into water.

**In vitro bioimaging**. The MIM structure on Si was attached to a glass slide with a double-sided tape. Then, a solution of PVA (Carolina Biological Supply Company, molecular weight 45 g/mol) was poured onto the MIM structure so that the PVA flooded the entire Si substrate. The PVA was then left to dry in air overnight at room temperature. Then, the portion of the PVA film covering the MIM structure was excised with a razor blade and tweezers. The PVA film containing the MIM structures were dissolved in 200 μL of deionized water at ~70 °C until the film dissolved completely in water. The solution was then dropcast onto a Cu TEM grid with Carbon backing (Ted Pella) for TEM imaging. The TEM images were acquired with Phillips CM100 TEM with the operating voltage of 80 kV.

For in vitro imaging, 100 μL of the MIMs dispersed in water was incubated with T24T bladder cancer cells (American Type Culture Collection) for an hour. The cells were imaged with Zeiss LSM780 that is equipped with confocal upconversion fluorescence imaging using a tunable NIR Ti:Sapphire laser (Coherent Chameleon Ultra II).

**Photoluminescence characterization**. The geometric features of the MIM nanostructures were characterized with a field-emission scanning electron microscope (JEOL 7401F). The PL spectroscopy was conducted using a confocal laser scanning microscope coupled with a spectrometer (Renishaw InVia). The samples were excited with a 980 nm wavelength laser (CrystaLaser DL980-500) and the emission from the UCNPs green channel (500–600 nm) was collected for analysis.

In the transient PL measurements, an excitation laser source (Thorlabs L980P200) was modulated by a square pulse generated by a function generator (Wavetek model 166). The duty ratio, pulse duration, and voltage amplitude were set appropriately, so that the transient PL had enough time to reach the steady state. Transient PL measurements were performed for both the NIR and green luminescence. The emitted PL from samples was collected by two convex lenses before being focused into a monochromator (Sciencetech 9057F) equipped with two photomultiplier tubes (PMTs), one for the visible (Hamamatsu H11461P-11) and the other for the NIR detection (Hamamatsu H10330B-75). Finally, a photon counter (Stanford Research Systems SR430) was used to convert the PMT output to digital transient waveforms.

**Code availability**. All simulation results presented here are generated using the commercial finite element time domain software Lumerical (version 2018a). Relevant simulation files and post-processing scripts are available from authors upon request.

## Data availability

All relevant data are available from the corresponding author upon reasonable request.

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

## Acknowledgements

This work was supported in part by the National Science Foundation through Grant DMR-1420736, MRSEC: Soft Materials Research Centre, the Army Research Office through Grants W911NF-14-1-0211 and W911NF-14-1-0463, the National Research Foundation (NRF), Ministry of Information and Communication Technologies (ICT) and Future Planning of Korea (NRF-2016K1A1A2912758), and the Colorado Office of Economic Development & International Trade (CTGG1 2017-0609). The authors thank Kyuyoung Bae for helpful discussion. We also thank Nikki Rentz and Kimberly Briggman for their assistance with the DLS measurements.

## Author contributions

A.D. conducted electrodynamic simulations, fabrication of MIM structures and PL measurements. C.M. conducted time-resolved PL and contributed to rate equations analysis. S.C. synthesized UNCPs and also helped with the fabrication. K.K. contributed to the idea generation and analysis. W.P. conceived the idea, led all aspects of the research, and manuscript preparation.

## Additional information

**Competing interests:** The authors declare no competing interests.

