## [Peer Review File · Nature Communications]

Reviewers' comments:

Reviewer #1 (Remarks to the Author):

The authors present a metal upconverter sandwich structure for which they report high upconversion luminescence enhancement. The structure is innovative and the paper is generally well written. However, there are some open questions, which must be clarified prior to publication and some overselling in the conclusion /discussion part which should be avoided.

The structure the authors present has a very strong angular characteristic. It appears that the samples were measured under ideal excitation conditions that is the field enhancement is at its maximum. Therefore, we must assume that in other angular directions the performance is much worse. There is a graph illustrating the strong decrease in performance in the supplementary information (SF 4). From its captions it appears that the authors have simply averaged the results for z and x excitation direction to come up with the number of 500 time PL enhancement. First, it is most certainly wrong to assume that the angular behavior can just be interpolated between the two cases. I can imagine many structures where this is not true, at least. Second, how was the PL enhancement derived from the absorption enhancement? Furthermore, only light emitted into the excitation direction is detected. In consequence, we do not know whether PL was really enhanced by the reported factors, but only that emission into that specific direction. Overall emission enhancement could be much lower.

How do the quenching factors determined from the time resolved measurements compare to the intensity measurements presented Figure 3? You predict absorption enhancement by a factor of 300. You determine a characteristic exponent in the low excitation regime of roughly 2. This would mean that a 300 fold absorption enhancement would result in a 90000 fold PL enhancement. Instead you determine only a roughly 1000 fold enhancement. This suggests very strong parasitic processes. There is also some confusion, because in Figure 3b one label reads "measured abs enhancement" and in Figure 3 d it is the PL enhancement. Is this correct (then how was the absorption determined) or a mistake?

The upconverter layers in the reference are thicker than in the plasmonic device. The paper states that the "PL intensities are calibrated to account for the difference in the number of UCNPs". How was this done? Please note that a linear scaling would most certainly be wrong. With more particles (i.e. in the reference) there is also more absorption such that some particles receive a lower intensity. Because of the non-linear nature of upconversion this has to be accounted for. Was only the different thicknesses being accounted for, or also the different coverage of the surface. I would object to the latter, because the structures might "suck" also some energy from their surroundings.

In the introduction and also to the end of the paper, the authors compare their results with literature, claiming that the achieved enhancement are well above results previously reported. For a fair comparison the authors should answer the question, at which irradiance the other enhancement factors were reported. As visible from Figure 3d this makes a huge difference and only comparisons at the same irradiance level are meaningful. Furthermore, following up on the discussion of angular behavior above, it is also important whether these enhancements were determined in one emission direction only (like in the present paper) or perhaps with an integrating sphere setup, as well as whether the samples were "ordered" (like in the present paper) or already in a solution (like in the intended application).

Even by your own analysis you cannot fully explain the differences in the enhancement factors (168 to 250), so I would not claim in the conclusion that you have excellent agreement between theory and experiment. Furthermore, you do not have achieved precise fabrication of the intended structure as you show yourself with the SEM images. Finally, following up the discussions above, I assume that the real enhancement factors relevant in any application will be much lower. So

overall I would formulate more cautious in the conclusion.

Reviewer #2 (Remarks to the Author):

The article "Over 1000-fold enhancement of upconversion luminescence of NaYF₄:Yb³⁺,Er³⁺ nanocrystals using water-dispersible plasmonic metal-insulator-metal nanostructures" by Das and colleagues, describes a new way to improve the brightness of UCNPs by using plasmonic enhancement. This is a very interesting approach even if the 1000-fold enhancement is difficult to compare to other UCNPs without enhancement. For example, the reference layer is much thicker than the MIM sample. The luminescence measurements seem to have been performed in a dry state. This should be described in the manuscript. How is the intensity difference in water dispersion? This is much more interesting. The authors should also try to determine the enhancement factor in suspension. In this case, they should think about using a different reference material. Finally, with a 1000-fold enhancement, each MIM should be individually visible under the fluorescence microscope equipped with a 980-nm excitation source and using a high NA objective lens. Such a microscopic measurement will enable the authors to determine the variation of brightness between individual MIMs. Regarding the cell imaging, the caption of Supporting Fig. 3 is unclear because there are different terms used: MIM vs. gold nanorods. If there are only gold nanorods, it is now wonder that the upconversion luminescence is almost non-existent. Under (c), the term UCNP-AuNR is used. Is this the MIM? Or is it a mixture of gold nanorods and single UCNPs?

Minor point:

Line 62: Citation is missing

Fig. 1: The definition of scale bars is missing

Line 375: There are different types of PVA. What is the MW of PVA used for these experiments?

Reviewer #3 (Remarks to the Author):

The authors report on an interesting strategy to prepare colloidal nanohybrids which consist of upconversion nanoparticles sandwiched between two gold layers of hundreds of nanometers. The procedure is based on lithographic fabrication and the use of a substrate exhibiting poor adhesion to gold and makes it possible to precisely control the orientation and distance of the plasmonic nanoparticles with respect to the UCNPs. The preparation of these colloids has been sufficiently described to reproduce the work.

A highly relevant outcome is the remarkable enhancement of the UCNP photoluminescence compared to those reported in the literature as well as the preliminary studies on their potential for biomedical applications, by using the colloids in cancer-cell imaging.

In my opinion, these results are novel, relevant, and of interest to several research fields, such as biology (in particular, bioimaging). However, the authors should perform a study in-depth of these new hybrids:

- The nature of the molecules at the nanohybrid surface that make them dispersible
- Fully physical/chemical characterization of these new hybrids: TGA, DLS, XPS,...
- Studies of their colloidal stability
- Analysis of their chemical stability

All these data are relevant for the application of these hybrids in biological systems.

Reviewer #1 (Remarks to the Author):

1. The authors present a metal upconverter sandwich structure for which they report high upconversion luminescence enhancement. The structure is innovative and the paper is generally well written. However, there are some open questions, which must be clarified prior to publication and some overselling in the conclusion /discussion part which should be avoided.

The structure the authors present has a very strong angular characteristic. It appears that the samples were measured under ideal excitation conditions that is the field enhancement is at its maximum. Therefore, we must assume that in other angular directions the performance is much worse. There is a graph illustrating the strong decrease in performance in the supplementary information (SF 4). From its captions it appears that the authors have simply averaged the results for z and x excitation direction to come up with the number of 500 time PL enhancement. First, it is most certainly wrong to assume that the angular behavior can just be interpolated between the two cases. I can imagine many structures where this is not true, at least.

It is true that our MIM structure exhibits maximum enhancement under excitation along the z-direction. Heuristically, if there is no enhancement for excitation along the xy plane, one should expect the overall enhancement to be a third of the maximum enhancement under excitation along the z direction. Given that there is still some enhancement from excitation along the xy plane for the p polarization (See response to Reviewer 2 Comment 3), the overall enhancement should be slightly larger than a third.

In order to fully characterize the angular dependence of the MIM enhancement, we conducted further simulations, sweeping the incident angle in increments of 9 degrees for both p and s polarizations. Using these results, we estimated the average absorption enhancement with all polarizations and incident angles accounted for. For an MIM with a predicted enhancement of 275 for excitation along the z axis, we found an average enhancement of 112 which is slightly larger than 1/3 of the original value (about 41%), as expected.

In our original manuscript, we extrapolated the simulated absorption enhancement to predict a weak PL enhancement of 500 for our dispersed MIMs. We were unable to actually measure the weak PL enhancement in solution as our sample did not have a high enough concentration of

MIMs. Given this, in our revised manuscript, we have removed statements regarding the predicted weak PL enhancement as we feel they may be misleading to the reader. Instead, we simply state the enhancement reduction of 41% that is suggested by our simulations.

In order to clarify the angular dependence, we have updated the relevant figure in the supplementary information and rewritten the corresponding caption as follows:

Supplementary Figure 4. Simulated enhancement MIM versus incident angle. a. Absorption enhancement as a function of incident angle θ measured from the z axis for p polarization (E field along xz plane). E field for the MIM plasmonic mode is primarily in the z direction so there is still enhancement for excitation along the x axis due to mode overlap (Incident E field at $\theta = 90^\circ$ points along z axis) **b.** Absorption enhancement as a function of incident angle θ measured from the z axis for s polarization (E field tangential to xy plane). Enhancement for excitation along the x axis is 0 for this polarization due to poor mode overlap. Averaging over all angles and polarizations, we calculate an expected enhancement of 112 which is 41% of the enhancement predicted for normal incidence.

2. Second, how was the PL enhancement derived from the absorption enhancement?

We used simulations to calculate the reduction in absorption enhancement for the dispersed versus normal incidence scenarios, which was then used to predict the reduction in PL enhancement from the measured normal incidence value. In the strong excitation regime, the PL scales linearly with incident power (Equation 4). Based on the simulation results of the expected absorption enhancement after averaging over all excitation conditions (see Supplementary Fig. 4), we expect the dispersed PL enhancement factor to be 41% of the PL enhancement factor

measured on the Si wafer. This was validated by our measured PL enhancement of the dispersed MIM in the strong excitation regime (see Reviewer 2, Comment 3).

3. Furthermore, only light emitted into the excitation direction is detected. In consequence, we do not know whether PL was really enhanced by the reported factors, but only that emission into that specific direction. Overall emission enhancement could be much lower.

Our PL measurements were performed with a 50x objective lens collecting signal over a range of emission angles. To confirm that our normal incidence enhancement measurements are representative of overall emission enhancement, we performed a comparison measurement using an integrating sphere. An enhancement of 255 was measured at a power density of 13 kW/cm². As seen in Figure 3d in the main text, we measured an enhancement of 240 at the same power density, confirming that the PL enhancements measured by our confocal microscope setup are representative of the overall enhancement.

4. How do the quenching factors determined from the time resolved measurements compare to the intensity measurements presented Figure 3? You predict absorption enhancement by a factor of 300. You determine a characteristic exponent in the low excitation regime of roughly 2. This would mean that a 300 fold absorption enhancement would result in a 90000 fold PL enhancement. Instead you determine only a roughly 1000 fold enhancement. This suggests very strong parasitic processes.

As the reviewer has pointed out, there is indeed non-radiative quenching by metal. Specifically, there are two quenching processes we should consider: quenching of the green emission and quenching of the intermediate energy levels in the NIR. Regarding how these quenching factors are related to the measured PL enhancements, this is a major part of our Discussion section with Equations 4 and 5 presenting the exact relationship. The first plateau in Figure 3d (at power densities above 40 kW/cm²) corresponds to the strong excitation regime, where both the MIM and reference PL scale linearly with power and Equation 4 is the relevant relationship to consider. The second plateau (at power densities below 0.03 kW/cm²) corresponds to the weak excitation regime, where both the MIM and reference PL scale quadratically with power and Equation 5 is the relevant relationship to consider. Note that while the effect of quenching on PL in the strong excitation regime (Equation 4) is relatively easy to see, the effect of quenching in the weak excitation regime (Equation 5) is nontrivial. This actually is one of the major findings this paper reports. Thus, the non-radiative quenching by the metal at green and NIR wavelengths prevents us from achieving over 4 orders of magnitude of enhancement in the weak excitation regime that would have been possible if there were no quenching.

We have added the following text to page 13 of the revised manuscript to emphasize the importance of accounting for the green and NIR quenching in our structures:

These quenching factors prevent us from achieving a simple quadratic scaling of enhancement in the weak excitation regime and must be accounted for with time resolved PL measurements.

5. There is also some confusion, because in Figure 3b one label reads “measured abs enhancement” and in Figure 3 d it is the PL enhancement. Is this correct (then how was the absorption determined) or a mistake?

We thank the reviewer for catching this problem and have fixed the 3b right y axis label to read ‘measured PL enhancement’ in the revised manuscript.

6. The upconverter layers in the reference are thicker than in the plasmonic device. The paper states that the “PL intensities are calibrated to account for the difference in the number of UCNPs”. How was this done? Please note that a linear scaling would most certainly be wrong. With more particles (i.e. in the reference) there is also more absorption such that some particles receive a lower intensity. Because of the non-linear nature of upconversion this has to be accounted for. Was only the different thicknesses being accounted for, or also the different coverage of the surface. I would object to the latter, because the structures might “suck” also some energy from their surroundings.

In our study, we assumed a linear scaling in thickness and also accounted for the different surface area covered of our MIM sample compared to the reference in our calibration. A 90nm thick reference was used in order to measure the enhancement in the weak excitation regime more reliably. While the MIM sample was still easily measurable at all power densities, our 30nm thick reference sample was not measurable at the lowest power densities. We switched to a 90nm thick reference sample after confirming that the 90nm sample emitted 3 times as much PL as the 30nm reference. This scaling was confirmed over various incident power densities, covering both the strong and weak excitation regimes.

We believe that a linear scaling is appropriate in this case due to the relatively small thickness of both reference samples considered. It is true that for a thick slab of an absorbing material, the particles deeper within the material will receive a lower excitation intensity. However, in our case, due to the small absorption cross section of our UCNPs (of the order of 10^{-20} cm²: see, for example, Page et al., *J. Opt. Soc. Am.* **15**, 996 (1998)), we believe a negligible amount of power is lost over the 90nm compared to the 30nm sample and thus a linear scaling is safe to assume. With an absorption coefficient of 6 cm⁻¹ (Lu et al, *ACS Nano* **8**, 7780–7792 (2014)), less than 0.006% of the original power is lost over a slab 100nm thick.

Similarly, the low absorption cross section also means that the nanoparticles are sensitive only to their immediate local field intensity (i.e. no “sucking” effect) and thus it is appropriate to assume a linear scaling with surface area coverage as well. The MIM structure does “suck” the field in, effectively increasing the absorption cross section. This is the essence of the plasmon

enhancement of absorption, which allows us to absorb more light with a smaller volume (or smaller number) of UCNPs.

To justify our use of a linear scaling, we have added the following text to page 8 of the revised manuscript:

We assume a linear scaling in our calibration, which is valid due to the small absorption cross section of the UCNPs³².

7. In the introduction and also to the end of the paper, the authors compare their results with literature, claiming that the achieved enhancement are well above results previously reported. For a fair comparison the authors should answer the question, at which irradiance the other enhancement factors were reported. As visible from Figure 3d this makes a huge difference and only comparisons at the same irradiance level are meaningful. Furthermore, following up on the discussion of angular behavior above, it is also important whether these enhancements were determined in one emission direction only (like in the present paper) or perhaps with an integrating sphere setup, as well as whether the samples were “ordered” (like in the present paper) or already in a solution (like in the intended application).

In the manuscript we compare our results to the lithographic enhancement achieved by Saboktakin et al (35x enhancement) and Zhang et al (100x enhancement). The 35-fold enhancement reported by Saboktakin et al was measured with a power density of 0.005 kW/cm² which falls well within the weak excitation regime. The 100-fold enhancement reported by Zhang et al was measured with a power density of 0.3kW/cm² which, according to their measurements, falls under the weak excitation regime. Thus, it is fair to compare our over 1000-fold enhancement to their results. While Saboktakin et al do not describe the angular setup of their measurements, the larger enhancement reported by Zhang et al was achieved using a setup similar to ours involving normal incidence and collection.

We also report the solution-based enhancement of 12.4 measured by Song et al. We were unable to find any information on the power density used to achieve the enhancement of 12.4 in the solution based measurement, however, even if the measurement was done in the strong excitation regime, our measurements show that we can achieve at least 4 times as much enhancement from our dispersed MIMs.

We have added the following text to page 16 of the revised manuscript:

The dispersed MIM structures reported in this paper show 4 times higher enhancement than this at high power densities and an order of magnitude improvement can be achieved by going to lower power densities.

8. Even by your own analysis you cannot fully explain the differences in the enhancement factors (168 to 250), so I would not claim in the conclusion that you have excellent agreement between theory and experiment. Furthermore, you do not have achieved precise fabrication of the intended structure as you show yourself with the SEM images. Finally, following up the discussions above, I assume that the real enhancement factors relevant in any application will be much lower. So overall I would formulate more cautious in the conclusion.

The manuscript was revised to be more conservative in the presentation of our results. Specifically, the following changes were made:

Line 85: Removed “ and excellent agreement between theory and experiments was demonstrated”

Line 159: Removed “extremely”

Line 297: Removed “The excellent agreement between measured values and theoretical analysis attests the high quality of MIM structures fabricated according to the design”

Line 339: Removed “excellent”

Line 370: Replaced “The excellent agreement between the theoretical analysis and experimental results demonstrate the MIM structures were fabricated precisely with the prescribe geometry with “The extremely high measured enhancement factors were found to be consistent with predictions from a theoretical analysis of the UCNP rate equations and electrodynamic modeling of the structure.

Reviewer #2 (Remarks to the Author):

1. The article “Over 1000-fold enhancement of upconversion luminescence of NaYF₄:Yb³⁺,Er³⁺ nanocrystals using water-dispersible plasmonic metal-insulator-metal nanostructures” by Das and colleagues, describes a new way to improve the brightness of UCNPs by using plasmonic enhancement. This is a very interesting approach even if the 1000-fold enhancement is difficult to compare to other UCNPs without enhancement.

For example, the reference layer is much thicker than the MIM sample.

Please see the response to Reviewer 1 Comment 6.

2. The luminescence measurements seem to have been performed in a dry state. This should be described in the manuscript.

We have added the following texts on page 9 and page 10 of the revised manuscript to explicitly state that the measurements shown in Figures 3 and 4 were performed in the dry state:

Page 8: Figure 3 shows the results from the PL measurements performed on the MIMs while still attached to the silicon substrate.

Page 9:

We also performed time resolved PL measurements on the MIM structures before lifting off and dispersing in water.

Page 14: Briefly summarizing our analysis of the measurements performed on the MIMs still on the silicon substrate

3. How is the intensity difference in water dispersion? This is much more interesting. The authors should also try to determine the enhancement factor in suspension. In this case, they should think about using a different reference material.

Based on the simulations presented in the new Supplemental Figure 4, we expect a reduction in enhancement when the MIMs are dispersed in water as they will be excited with light incident from all angles and polarizations. Using cylindrical coordinates to describe the structure, power incident along the radial axis with s polarization (E field tangential to the xy plane) will not be enhanced due to poor spatial overlap with the plasmonic mode while power incident along the radial axis with p polarization (E field along z axis) will show some enhancement. Thus, as explained in detail in our response to Reviewer 1 Comment 1, we expect the dispersed MIMs to have an enhancement slightly larger than 1/3 (41%, to be exact) of the enhancement measured with normal incidence excitation.

This was confirmed by measuring the PL enhancement of MIMs dispersed in water, which showed an enhancement of 50 when dispersed (about 43% of original value of 117 measured on the substrate). This agrees well with predictions from simulations. This measurement was done in the strong excitation regime. Even larger enhancement should be achievable at lower power densities.

We have made the following changes to include the results of our measurement of dispersed enhancement:

Page 4: The MIMs were dispersed into water and large enhancement in solution was demonstrated.

Page 10: Finally, the enhancement in PL by the MIMs in solution was confirmed by dispersing the MIMs into water. The MIM sample was compared to a reference of UCNPs in solution and the measured PL was calibrated to account for the difference in concentrations of the two samples. The dispersed MIMs had an enhancement factor of 50 in the strong excitation regime. We could not measure the enhancement in the weak excitation regime as the concentration of MIMs was not high enough to produce a measurable signal.

Page 14: To predict the expected enhancement factor of the MIMs once dispersed, we simulated the structure under all possible excitation angles and polarizations. Due to the cylindrical symmetry of the structure, we only have to consider excitation along the z axis and excitation at two polarizations along the radial direction. We found that, in addition to the large enhancement demonstrated by excitation along the z axis, enhancement also occurs for one of the two polarizations along the radial direction (see Supplementary Fig. 4). Averaging over all angles and polarizations, we predicted an enhancement that is about 41% of the enhancement measured from the structures on the silicon substrate. The dispersed MIMs had an enhancement factor of 50 in the strong excitation regime which is 43% of the enhancement factor of 117 that was measured for the structures still on the substrate and agrees well with predictions. As seen in Figure 3d, this enhancement should become even larger in the weak excitation regime where the nonlinear dependence on incident power density can be exploited.

4. Finally, with a 1000-fold enhancement, each MIM should be individually visible under the fluorescence microscope equipped with a 980-nm excitation source and using a high NA objective lens. Such a microscopic measurement will enable the authors to determine the variation of brightness between individual MIMs.

All the results presented in the original manuscript were done using a confocal set up with a spot size of 4.5 μm . This corresponds to an area which contains around 30 MIMs. To quantify the variation in PL from individual MIMs within this area, we performed a scan over a grid of MIMs with a higher NA objective lens. The figure below shows the PL results from the sample. As seen from the image, each MIM can be individually resolved. PL variation is within 15 percent of the mean.

We have added the figure above to Supplementary Figure 1 and adjusted the figure caption to:

Supplementary Figure 1. Uniformity of MIM sample. **a** SEM image of MIM sample confirming even deposition and clean resist lift off. Image contains roughly 1000 MIMs. **b** PL from individual MIMs collected with high NA objective lens. Individual PL was collected over an area corresponding to the spot size of excitation for the results presented in the main text. PL variation is within 15 percent of average PL. **c.** Percent variation from PL intensity averaged over 30 MIMs taken over a $300\ \mu\text{m} \times 300\ \mu\text{m}$ square of MIMs with $10\ \mu\text{m}$ step size. PL variation is within 10 percent of average value over roughly 100000 MIMs. Scale bar: **a** $1\ \mu\text{m}$.

5. Regarding the cell imaging, the caption of Supporting Fig. 3 is unclear because there are different terms used: MIM vs. gold nanorods. If there are only gold nanorods, it is now wonder that the upconversion luminescence is almost non-existent. Under (c), the term UCNP-AuNR is used. Is this the MIM? Or is it a mixture of gold nanorods and single UCNPs?

In Supplementary Figure 3, we compare the MIMs to a solution of UCNP-gold nanorod (AuNR) clusters developed in another project. Through a modified PEGylation process, individual UCNPs are attached to gold nanorods that support a plasmon resonance at 800nm. This UCNP-AuNR nanocluster was shown to exhibit the same PL intensity as individual UCNP thanks to the deliberate detuning of the plasmon resonance away from the emission and absorption wavelengths of UCNP (see figure below). Thus, the PL from the UCNP-AuNR nanocluster serves as a qualitative reference for PL from bare UCNPs that are not plasmonically enhanced. By doing this, we did not have to produce water-soluble UCNPs separately.

Supplementary Figure 3c shows the PL from the UCNP-AuNR clusters as a comparison to the PL from the MIMs presented in this paper (shown in Supplementary Figure 3d). By comparing the two figures we can see that similar levels of brightness are achieved from a much lower concentration of MIMs. It is noted that this is a qualitative comparison designed to show the potential of our MIM structures for bioimaging applications. A more thorough study on bioconjugation and bioimaging with MIMs will be reported in our future publication.

We have changed Supplementary Figure 3a to show the PL from a single UCNP-AuNR cluster vs the PL from a single UCNP that demonstrates the validity of using the in-vitro UCNP-AuNR PL as a reference.

We have also edited the caption as follows to make the comparison clearer:

Supplementary Figure 3. MIM versus UCNP - gold nanorod cluster comparison. a. PL from a UCNP conjugated to a gold nanorod (AuNR) that supports a plasmon resonance at 800nm vs bare UCNP. Due to the deliberate detuning of the plasmon resonance away from the emission and absorption wavelengths of UCNP, PL from UCNP-AuNR cluster is similar to PL from bare UCNP and thus can be used as a reference to evaluate in-vitro MIM performance. **b.** Overlay image of brightfield image with the green upconversion photoluminescence image of T24T cells incubated with UCNP-AuNR nanoclusters. **c.** A zoomed in image of the UCNP-AuNR nanocluster upconverted PL image in b and **d.** a zoomed-in image of the MIM upconversion PL image in Figure 5c. After accounting for the acquisition conditions (irradiation power and detector gain), similar levels of brightness were obtained with a 3 orders of magnitude lower concentration of MIMs than UCNP-AuNR clusters. Scale bars: **a** 100 nm; **b** 20 μm

6. Line 62: Citation is missing

Added citation to line 62.

7. Fig. 1: The definition of scale bars is missing

We have added the following text to page 5 of the revised manuscript:

Figure 1c shows the simulated field profile of the MIM structure normalized by the incident plane wave excitation.

We have also altered the caption in Figure 1c to:

c Simulated field profile under 980 nm normal incidence plane wave excitation. E field is normalized by incident plane wave amplitude E_0 .

8. Line 375: There are different types of PVA. What is the MW of PVA used for these experiments?

According to the supplier of the PVA, the molecular weight of the PVA we used was 45 g/mol. The liftoff process we have used is a purely physical process relying on the adhesive properties of PVA and should not be sensitive to the molecular weight of the PVA.

We have included the item number in our methods section in case someone wants to exactly replicate our results.

Reviewer #3 (Remarks to the Author):

The authors report on an interesting strategy to prepare colloidal nanohybrids which consist of upconversion nanoparticles sandwiched between two gold layers of hundreds of nanometers. The procedure is based on lithographic fabrication and the use of a substrate exhibiting poor adhesion to gold and makes it possible to precisely control the orientation and distance of the plasmonic nanoparticles with respect to the UCNPs. The preparation of these colloids has been sufficiently described to reproduce the work.

A highly relevant outcome is the remarkable enhancement of the UCNPs photoluminescence compared to those reported in the literature as well as the preliminary studies on their potential for biomedical applications, by using the colloids in cancer-cell imaging.

In my opinion, these results are novel, relevant, and of interest to several research fields, such as biology (in particular, bioimaging). However, the authors should perform a study in-depth of these new hybrids:

1. The nature of the molecules at the nanohybrid surface that make them dispersible

The as-created MIM structures do not have any functional groups or charges on the surface that allows dispersion in water for a long period of time. However, the MIMs maintain structural integrity for up to 12 hours once dispersed in water (data presented below) which is enough time for additional coating techniques to be applied that will allow long-term stability in water.

We have added the following text to page 15 of the revised manuscript to address this:

The as-created MIM structures do not have any functional groups or charges on the surface that allows dispersion in water for a long period of time. Based on our TEM and DLS studies, it can be seen that the MIM structures are stable in water for at least 12 hours under varying pH conditions. This grants sufficient time for subsequent surface modification for high stability. For example, the MIM structures can be coated with monofunctional thiol-polyethylene glycol (mPEG-SH) that will allow long-term colloidal stability in water. For improved structural integrity, a silica coating can be applied. These coating processes are typically overnight reaction processes⁵¹ and thus we believe they can be done with high efficiency.

2. Fully physical/chemical characterization of these new hybrids: TGA, DLS, XPS,..

We were unable to get a large enough quantity of MIM sample for the TGA measurement so the TGA was only performed on UCNPs that form the insulator layer of our MIM nanostructure. We believe the TGA results on UCNPs should be representative of any physical phenomena occurring in the MIMs at temperatures below 500°C since we do not expect to see any

degradation of gold in this temperature range. The plot below shows the results of the TGA on the UCNPs. From the TGA measurement, we see two major weight loss events occurring at around 150°C and around 350°C. We attribute these events to the evaporation of the residual 1 Octadecene alkene (flash point 155°C) and Oleic acid (boiling point 350°C) that are known to coat the UCNPs in our synthesis process.

Note that the UCNPs are stable up to 100°C and thus, we expect the MIMs to be as well. As the MIMs are intended primarily for bioimaging purposes, 100°C is well above any temperature conditions we expect the MIMs to be exposed to.

DLS results are presented in the response to the next comment. We do not have access to XPS.

We have added the figure above to our supplementary information along with the following caption:

Supplementary Figure 6. Thermal stability of UCNPs. Thermogravimetric analysis of UCNPs shows that the MIMs are stable for temperatures up to 100°C. We observed two major weight loss events occurring at around 150°C and around 350°C. We attribute these events to the evaporation of the residual 1 Octadecene alkene (flash point 155°C) and Oleic acid (boiling point 350°C) that are known to coat the UCNPs in our synthesis process.

We have also added the following text to the revised manuscript:

Page 4: In addition to this, physical and chemical stability studies of the dispersed MIMs were performed to further demonstrate the bio-applicability of the nanostructures.

Page 15: We also studied the physical, chemical and thermal stability of the dispersed MIM structures (see Supplementary Figs. 5, 6).

3. Studies of their colloidal stability - Analysis of their chemical stability

All these data are relevant for the application of these hybrids in biological systems.

We studied the colloidal and chemical stability of the dispersed MIMs by monitoring the results from DLS measurements over a period of 24 hours. To gauge chemical stability, the MIMs were dispersed into buffer solutions with pHs of 4,7 and 10. All samples were stable up to 8 hours. The measurement made at the 24 hour mark could not be analyzed using DLS as the sample had become too polydisperse, suggesting disintegration of the MIMs into their constituent layers. This was confirmed from TEM images of the samples. TEM images also confirmed stability of the MIMs for up to 12 hours once dispersed into water. As discussed above, this is enough time to further functionalize the nanostructures to increase their stability in water.

We have added the following text to page 15 of the revised manuscript:

The colloidal stability of the MIM nanostructures in solution was investigated by monitoring dynamic light scattering (DLS) measurements of the MIM solution over time. Sensitivity to the solutions pH value was also tested by preparing three buffer solutions with pH values of 4,7 and 10. The DLS measurements show that the particles are stable for up to 8 hours in all three buffer solutions (see Supplementary Fig. 5). A final measurement was performed 24 hours after dispersal but by this point the sample was too polydisperse to perform DLS analysis suggesting the MIMs had broken apart into their constituent layers. This was confirmed with TEM images of the dispersed samples which showed structural integrity up to 12 hours and degradation at 24 hours.

The as-created MIM structures do not have any functional groups or charges on the surface that allows dispersion in water for a long period of time. Based on our TEM and DLS studies, it can be seen that the MIM structures are stable in water for at least 12 hours under varying pH conditions. This grants sufficient time for subsequent surface modification for high stability. For example, the MIM structures can be coated with monofunctional thiol-polyethylene glycol (mPEG-SH) that will allow long-term colloidal stability in water. For improved structural integrity, a silica coating can be applied. These coating processes are typically overnight reaction processes⁵¹ and thus we believe they can be done with high efficiency.

We also added an additional Supplementary Figure showing the results of the DLS measurements:

Supplementary Figure 5. Colloidal stability over time. **a** Physical and chemical stability of dispersed MIMs was tested by monitoring dynamic light scattering measurements of various buffer solutions over a 24 hour period. The samples were stable over 8 hours but deteriorated by the 24 hour mark. **b** TEM image confirms colloidal stability up to 12 hours **c** TEM image showing degradation at 24 hours. Scale bars: **b** 500 nm; **c** 100 nm

Reviewers' comments:

Reviewer #1 (Remarks to the Author):

The authors have extensively revised their manuscript and explained many open questions.

There is only one point left, where I disagree with their methodology. They have clarified that they do account for the difference in the number of UC particles by a) linearly scaling with the thickness and b) considering the reduced area of the their plasmonic structure. With point a) I can live given the low overall absorption. Part b) however, is misleading as the structure is able to harvest energy from its surroundings. In any application the relevant question will be "how much more PL signal" will I get back, when I use the MIM structure. Most likely in all this applications, the area to apply the UC structure to, will be limited. Because of the harvesting effect, I will not be able to put the MIM structures close to each other without loosing some enhancement. In consequence, I will not get the PL signal I would get from the enhancement reported in the paper, but a lower one. Therefore, I strongly suggest that the authors do not use the correction for area, but only for the thickness.

Reviewer #2 (Remarks to the Author):

The authors have addressed all questions of the reviewers.

Suppl. Figure 1b is an important addition. Instead of "high NA" objective, the authors should directly provide the NA. "15 % variation" is too ambiguous and surprisingly small for the difference in the intensity seen between individual MIMs. The authors should add another panel to Fig. 1, plot the integrated PL intensity of a statistically relevant number of individual MIMs as a histogram, apply a Gaussian fit to the histogram and determine the standard deviation from the Gaussian fit. This will give a much better picture of the PL heterogeneity in the sample. The coefficient of variation is likely higher than 15 %.

The authors should make sure that the experiments are reproducible by others. Thus, they should also include the MW of 45 g/mol in the methods section.

Reviewer #3 (Remarks to the Author):

The authors have made satisfactory revisions to the manuscript in response to my previous comments. In addition, they have addressed most of the other reviewers' comments. Overall the manuscript reads well and, in my opinion, is suitable for publication in Nature Communications.

Reviewer #1 (Remarks to the Author):

The authors have extensively revised their manuscript and explained many open questions.

There is only one point left, where I disagree with their methodology. They have clarified that they do account for the difference in the number of UC particles by a) linearly scaling with the thickness and b) considering the reduced area of the their plasmonic structure. With point a) I can live given the low overall absorption. Part b) however, is misleading as the structure is able to harvest energy from its surroundings. In any application the relevant question will be "how much more PL signal" will I get back, when I use the MIM structure. Most likely in all these applications, the area to apply the UC structure to, will be limited. Because of the harvesting effect, I will not be able to put the MIM structures close to each other without losing some enhancement. In consequence, I will not get the PL signal I would get from the enhancement reported in the paper, but a lower one. Therefore, I strongly suggest that the authors do not use the correction for area, but only for the thickness.

In order to address the effects of neighboring MIMs on the plasmonic enhancement, we simulated the unpolarized normal incidence excitation of two MIMs as a function of the distance between them. The figure below shows the results of the simulations. For separations larger than 500 nm, the absorption enhancement in the two MIMs is similar to that of a single, isolated MIM. There are small variations occurring near integer multiples of the plasmonic wavelength (here 980 nm), which is due to the radiative interaction between the two MIMs, i.e. the interference of scattered waves. It is clear that the radiative interaction makes only small changes and the enhancement is predominantly due to the plasmon resonance of individual MIM structures.

Not accounting for the difference in surface area coverage between the reference and the MIM samples would result in a reported enhancement that is dependent on the periodicity of the sample. Consider, for example, two identical MIM samples made from hole arrays with two different periodicity, 500 nm and 1000 nm. Despite being identical, the former would have 4

times higher enhancement than the latter, simply because there are 4 times more MIMs per area. This is unphysical as the plasmonic mode is not dependent on the periodicity of the mold array.

Additionally, all the measurements reported in the paper were made from MIMs that were separated by distances greater than 500 nm where the enhancement of array sample is the same as that of an isolated single MIM.

Note also that the measured enhancement of the MIMs dispersed in solution was calibrated by the concentration of the sample versus a reference solution and thus has no calibration factor associated with surface area. The measured enhancement of the dispersed MIMs is consistent with the reduction expected from normal incidence measurements on MIMs that are not experiencing effects from neighboring MIMs.

Given all this, we believe it is fair and physically meaningful to report the enhancement measured after accounting for area coverage.

While we stand by our approach of calibration, we do recognize the reviewer's point that one might not get the same enhancement from a dense packing of MIMs. To address this, we will add the enhancement vs distance figure below in the SI.

We have added the figure above to the Supplementary Information of our manuscript along with the following caption:

Supplementary Figure 7. Simulated MIM enhancement versus separation distance. Simulated absorption enhancement of two MIMs under normal incidence unpolarized excitation as a function of the separation distance between them. For separations larger than 500 nm, the

absorption enhancement in the two MIMs is similar to that of a single, isolated MIM, with small variations occurring near integer multiples of the plasmonic wavelength (here 980 nm) due to the radiative interaction between the two MIMs.

We have also included the following text on page 16 in the main manuscript referencing the figure:

We also studied the effects of neighboring MIMs as a function of the spacing between the MIMs (see Supplementary Fig. 7). Our simulations show that separation distances larger than 500 nm are sufficient to avoid any reduction in enhancement due to nearby MIMs. Thus, our results demonstrate that MIMs can easily replace existing water-soluble UCNPs while drastically reducing the total number of nanoparticles needed for high quality cell imaging.

Reviewer #2 (Remarks to the Author):

The authors have addressed all questions of the reviewers.

Suppl. Figure 1b is an important addition. Instead of “high NA” objective, the authors should directly provide the NA.

The NA of the objective has been added to the caption of Supplementary Figure 1b. We have adjusted the caption as follows:

b PL from individual MIMs collected with high NA objective lens (NA 1.4).

“15 % variation” is too ambiguous and surprisingly small for the difference in the intensity seen between individual MIMs. The authors should add another panel to Fig. 1, plot the integrated PL intensity of a statistically relevant number of individual MIMs as a histogram, apply a Gaussian fit to the histogram and determine the standard deviation from the Gaussian fit. This will give a much better picture of the PL heterogeneity in the sample. The coefficient of variation is likely higher than 15 %.

To fully characterize the individual variation in PL, we performed a more rigorous statistical analysis on a larger number of MIMs. The PL from 2500 MIMs was measured and a Gaussian fit was applied to the histogram of the PL counts. The results are shown in the figure below. The Gaussian fit shows a coefficient of variation of 28%.

We have replaced Supplementary Figure 1b with the histogram of the 2500 MIM PL measurements along with an image of the scan performed in the inset of the figure. We have adjusted the figure caption as follows:

b PL from 2500 individual MIMs collected with high NA objective lens (NA 1.4). A Gaussian fit applied to the histogram of the individual counts shows a coefficient of variation of 28%.

The authors should make sure that the experiments are reproducible by others. Thus, they should also include the MW of 45 g/mol in the methods section.

The MW of the PVA has been added to the methods section. The corresponding sentence has been adjusted as follows:

Then, a solution of PVA (Carolina Biological Supply Company, molecular weight 45 g/mol) was poured onto the MIM structure so that the PVA flooded the entire Si substrate.

Reviewer #3 (Remarks to the Author):

The authors have made satisfactory revisions to the manuscript in response to my previous comments. In addition, they have addressed most of the other reviewers' comments. Overall the manuscript reads well and, in my opinion, is suitable for publication in Nature Communications.

We are pleased to hear the positive comments by the reviewer.

REVIEWERS' COMMENTS:

Reviewer #1 (Remarks to the Author):

I can live with the presentation of the distance dependence as substitute for omitting the scaling by area. Together with the added information the paper can now be published.